# GeoLOD: A Spatial Linked Data Catalog and Recommender

**Vasilis Kopsachilis** *,† and **Michail Vaitis** †

Department of Geography, University of the Aegean, GR-811 00 Mytilene, Greece; vaitis@aegean.gr
* Correspondence: vkopsachilis@geo.aegean.gr
† Current address: Department of Geography, University Hill, GR-811 00 Mytilene, Greece.

**Abstract:** The increasing availability of linked data poses new challenges for the identification and retrieval of the most appropriate data sources that meet user needs. Recent dataset catalogs and recommenders provide advanced methods that facilitate linked data search, but none exploits the spatial characteristics of datasets. In this paper, we present GeoLOD, a web catalog of spatial datasets and classes and a recommender for spatial datasets and classes possibly relevant for link discovery processes. GeoLOD Catalog parses, maintains and generates metadata about datasets and classes provided by SPARQL endpoints that contain georeferenced point instances. It offers text and map-based search functionality and dataset descriptions in GeoVoID, a spatial dataset metadata template that extends VoID. GeoLOD Recommender pre-computes and maintains, for all identified spatial classes in the Web of Data (WoD), ranked lists of classes relevant for link discovery. In addition, the on-the-fly Recommender allows users to define an uncatalogued SPARQL endpoint, a GeoJSON or a Shapefile and get class recommendations in real time. Furthermore, generated recommendations can be automatically exported in SILK and LIMES configuration files in order to be used for a link discovery task. In the results, we provide statistics about the status and potential connectivity of spatial datasets in the WoD, we assess the applicability of the recommender, and we present the outcome of a system usability study. GeoLOD is the first catalog that targets both linked data experts and geographic information systems professionals, exploits geographical characteristics of datasets and provides an exhaustive list of WoD spatial datasets and classes along with class recommendations for link discovery.

**Keywords:** linked data; spatial datasets; data catalog; dataset recommender





## 1. Introduction

Linked data principles [1] lay the technological background for data publishing on the web so that they can be transparently and uniformly accessed by humans and software. Link establishment among related data increases data sharing, interoperability, and reuse; aids dataset enrichment; and unleashes powerful retrieval capabilities already exploited by question answering [2–5] and query federation [6–11] systems. The idea of a web of open and interlinked data has been embraced by scientists and organizations, and steps have been taken towards this direction during the last decade or so. At the early stages of linked data development, providers such as DBpedia [12], MusicBrainz [13] and GeoNames [14] converted their data to RDF and made them accessible through dumps, SPARQL endpoints or embedded them in HTML documents using RDFa [15]. Since then, many tools have been developed, such as search engines [16,17], data catalogs [18,19], link discovery frameworks [20,21], and dataset recommenders [7,22–24], forming the linked data tools ecosystem and facilitating users to consume linked data and lowering the barriers for its adoption by non-expert users. Today, the Linked Open Data (LOD) cloud diagram includes more than 1200 datasets, and DataHub maintains metadata for more than 700 datasets. References [25,26] note that linked data size is expanding and the number of the LOD cloud diagram datasets increased from 203 to 1269 during the period 2010–2020. LODLaundromat [27] reports 38 billion indexed triples in 2018.

The increasing availability of linked data provides more options to users, but at the same time, increases the difficulty in identifying the appropriate data sources that meet their needs. Concerning linked data search, user needs vary and some common scenarios include searching for (a) topic-specific datasets (e.g., about conferences, music, or geography) [28]; (b) datasets that contain a given entity [29,30]; and (c) similar datasets to a given dataset [23,31]. These scenarios are being covered by available tools and applications; however, to the best of our knowledge, there is not a tool that addresses user needs related to geographical aspects of datasets during linked data search and exploration. In this work, we identify and address four possible scenarios:

1.  A user searches for datasets that cover a specific geographical area (e.g., a country);
2.  A linked data publisher searches for datasets containing georeferenced information in order to georeference their data;
3.  A linked data publisher searches for datasets that contain related instances to their own datasets in order to establish links between instances; and
4.  A geographical information systems (GIS) professional wants to enrich their spatial data with linked data.

These scenarios are covered in GeoLOD, a web catalog of spatial Web of Data (WoD) datasets and classes and a recommender for spatial datasets and classes that may contain related instances. The terms spatial datasets and spatial classes denote datasets and classes, respectively, that contain georeferenced instances, that is, instances whose locations are expressed with longitude and latitude coordinates. GeoLOD parses LOD cloud and DataHub catalogs, identifies spatial datasets and their spatial classes and extracts their metadata. It generates additional metadata that capture spatial aspects of datasets and classes, such as their bounding box and number of spatial entities and associated spatial vocabularies, and exposes them in GeoVoID, a vocabulary that extends the Vocabulary of Interlinked Datasets (VoID) [32], to describe spatial aspects of datasets. GeoLOD Catalog allows access to the lists of linked data spatial datasets and classes (along with their metadata) through a user interface and provides text and map-based search functionality, thus addressing scenarios 1 and 2.

GeoLOD Recommender generates ranked lists of spatial datasets and classes that may contain related instances with a given dataset or class, so as to be further examined in link discovery processes for the establishment of `owl:sameAs` links or other links that denote close semantic relation among their instances. The recommendation method is based on the work presented in [33] that builds a recommendation algorithm on the hypothesis that "pairs of classes whose instances present similar spatial distribution are more related than pairs of classes whose instances present dissimilar spatial distribution, in the sense that the former are more likely to contain semantically related instances" (p. 152), and thus are better candidates to be used as input in a link discovery process. GeoLOD applies the recommendation algorithm to generate recommendations for each class in the Catalog in the background. It allows the exploration for related classes and datasets through the user interface and the export of automatically generated SILK and LIMES configuration files for a selected pair of recommended classes that can be directly used for link discovery processes. Additionally, it allows on-the-fly recommendations for classes provided through a user-defined SPARQL endpoint, not listed in the Catalog, and for GeoJSON and Shapefile datasets, which are typical geographic information systems (GIS) file formats, thus addressing scenarios 3 and 4.

In addition to the user interface, GeoLOD provides a REST API to serve its content in well-known templates and formats, enabling software-based consumption. Specifically, it provides services that expose GeoLOD metadata and content description in the Data Catalog Vocabulary (DCAT) [34] format, an RDF vocabulary designed to facilitate interoperability among data catalogs, and dataset descriptions in GeoVoID that can aid source selection in query federation systems. It also provides services that export (a) SILK and LIMES configuration files for a selected pair of classes and (b) class recommendation lists

for datasets and classes in order to be consumed as input in batch link discovery processes. The main novelties of GeoLOD are:

- It is the first catalog of linked data spatial datasets and classes provided through SPARQL endpoints, offering services for describing spatial aspects of their content and map-based search;
- It introduces GeoVoID, an automatically generated dataset description vocabulary that extends VoID, to express spatial metadata and statistics of datasets;
- It provides a comprehensive list of recommended pairs of datasets and classes that may contain related instances, along with automatically generated SILK and LIMES configuration files and machine-readable recommendation lists so as to be used as input in (batch) link discovery processes; and
- It allows on-the-fly recommendations for user-defined SPARQL endpoints and spatial datasets in GeoJSON and Shapefile format.

The rest of the paper is organized as follows. In Section 2, we present applications related to linked data search and dataset recommendation. In Section 3, we present the design and methods of the GeoLOD application, and in Section 4, we present its implementation and the usage of the user interface and the REST API. In Section 5, we present statistics that summarize the linked data status regarding the geospatial domain, we assess the applicability of GeoLOD recommender in relation with the LIMES framework, and we evaluate GeoLOD usability by different user categories, namely linked data and GIS experts. We conclude the paper in Section 6 by discussing the results and by providing pointers for the improvement of the application.

## 2. Related Work

In this section, we present the work related to GeoLOD, classified into three categories: (a) vocabularies and tools for dataset description, (b) dataset catalogs, and (c) dataset recommenders for link discovery. We focus on prototypes and available systems for the linked data domain.

### 2.1. Dataset Description

VoID [32] (Vocabulary of Interlinked Datasets) is a well-known vocabulary for describing dataset content by expressing general information (such as, title, keywords, distribution URL, and provenance metadata), statistics (such as number of triples, classes, and properties), and connectivity details to other datasets. It aims to facilitate users and software agents in their dataset exploration [35], and many tools have been developed to generate automated VoID-based or similar dataset descriptions and statistics [36,37]. For example, RDFStats [38] provides an API that generates statistical items for SPARQL endpoints and RDF documents, including instance counts (per class) and histograms (per class, property and value type), originally developed to aid query federation systems. ExpLOD [39] summarizes RDF datasets usage and interlinking by computing representative dataset graphs and statistics, such as number of class instances and predicates used to describe an instance. LODStats [40] defines 32 statistical criteria, extending those defined in VoID, in a scalable and high-performance framework. Aether [41] is a statistics generator and visualization web application that focuses on comparing datasets between versions and on error detection. Loupe [42] and ABStat [43] produce ontology-driven dataset summaries that highlight their structure. ProLOD++ [44] augments dataset analytics with data mining functionality for identifying dependencies between dataset entities such as synonymously used predicates. In addition to dataset statistics, several tools, including LODex [45], LOD-Vader [46], LODAtlas [47], and LODSynthesis [31], provide high-level dataset summaries and visualizations. Concerning the description of geographical elements of the datasets, VoID supports the expression of their geographical coverage (e.g., bounding box) using the Dublin Core [48] spatial coverage predicate, and LODStats allows the (indirect) computation of geographical coverage by combining the minimum and maximum statistical criteria of longitude and latitude property values. Nevertheless, none of the above-described

vocabularies and tools capture the geographical aspects of datasets covered in this work, such as the number of georeferenced instances in datasets and classes.

### 2.2. Dataset Catalogs

Dataset catalogs provide single entry points for available linked-data datasets, and the most prominent examples are arguably the Linked Open Data (LOD) cloud and the DataHub. The LOD cloud visualizes datasets by topic, portrays their connectivity, and exports the list of its datasets in JSON format along with their basic provenance and descriptive dataset-level metadata, such as title, description, domain, point of contact, and distribution info (e.g., access URL and SPARQL enpoint URL). DataHub provides a user interface and a CKAN API (an API for querying data catalogs) for searching and filtering (not exclusively RDF) datasets and viewing their metadata. Both catalogs are populated through user-submitted datasets and metadata. LODAtlas [47] is a data catalog that provides keyword search and faceted navigation for RDF datasets parsed from several other catalogs including DataHub, Europeana, and Data.gov. It maintains dataset metadata, statistics about the number of their triples and their in- and out-going links. Moreover, it allows concurrent and in-depth exploration and comparison of multiple datasets' characteristics and provides an overview of their connectivity based on visual summaries. LODLaundromat [49] aims to improve linked data quality by republishing data in a "cleaner" state after correcting syntax errors, filtering duplicates, replacing blank nodes, etc. As part of the cleaning process, it offers description and search services for 650,000 cleaned RDF datasets (mostly data dumps). SPARQLES [50] monitors more than 500 SPARQL endpoints, collected from DataHub, regarding their availability, performance, interoperability, and discoverability, and provides a user interface for humans and an API for software agents for consuming its content. SPORTAL [51] is a catalog of SPARQL endpoints that allows SPARQL and keyword-based search. Endpoints are profiled by extended VoID descriptions, computed by directly querying their content. IDOL [52] provides metadata and analytics about an exhaustive list of RDF datasets in various formats (e.g., zip files and SPARQL endpoints), located by parsing eight data catalogs (including LOD cloud, LODLaundromaut, and the Registry of Research Data Repositories [53]). However, the list of datasets and their analytics are available only through a dump file. Contrary to the above generic data catalogs, LSLOD [54] and YummyData [55] are domain-specific data catalogs. The LSLOD Catalogue contains 52 life-sciences-related SPARQL endpoints for serving ontology alignment purposes between different datasets in the life science domain. Even though some catalogs allow (indirectly) the search for spatial datasets (e.g., in LODAtlas, users can retrieve spatial datasets by selecting a spatial vocabulary in the faceted search component), GeoLOD, to the best of our knowledge, is the first geographical domain data catalog that provides summaries for spatial aspects of datasets. Moreover, GeoLOD Catalog implements some novel features like the map-based dataset and class search and the on-the-fly projection of class spatial instances on an interactive map.

### 2.3. Dataset Recommenders for Link Discovery

Link Discovery refers to the problem of identifying and interlinking pairs of instances between two given triplesets for which a relation holds [56]. Two well-known link-discovery frameworks are SILK [20] and LIMES [21], which execute a link discovery process by allowing the set up of a workflow in configuration files or in user interfaces. The general workflow of a link discovery process consists of (a) providing as input two triplesets (e.g., two datasets or two classes), usually referred to as source and target, respectively; (b) defining the type of relation between their instances that will be discovered and established (e.g., `owl:sameAs`, which means the two instances refer to same real-world object); (c) defining the matching rule, that consists of one or more similarity metrics and the instance properties that will be evaluated (e.g., string equality of instance labels); and finally (d) executing the workflow to generate the recommended links between the instances of the two triplesets. A common obstacle for initiating a link discovery process is

that sometimes there is no prior knowledge of which two triplesets can be used as input for the link discovery process, or a linked data publisher may not be aware of target triplesets that are likely to contain related instances with their (source) tripleset. This is the focus of the Dataset Recommendation for Link Discovery domain, which refers to the automated process of recommending triplesets (e.g., datasets or classes) that may contain related instances to a given tripleset in order to be used as input in a link-discovery process.

Although several methodologies have been proposed to address the problem of Dataset Recommendation for Link Discovery [22,28,57–62], only few are implemented in tools and web applications [23,31,63,64]. One of them, the FluidOps portal [64], offers a data source exploration service, involving users in the source selection process, where a user begins to explore by providing some input (e.g., a keyword) and then refining the results through faceted search. It employs a data source contextualization method for discovering sources containing "somehow" related entities, and thus can serve link-discovery and distributed query processing tasks. TRT [63] recommends relevant triplesets for link discovery by applying link prediction metrics on a graph that maintains dataset connectivity information extracted from DataHub metadata. TRTML [23] augments the recommendation process with supervised learning algorithms. The input to the TRT/TRTML tool is the VoID description of the tripleset that the user wants to get recommendations, and the output is a ranked list of relevant triplesets for link discovery. LODSynthesis [31] is a suite of services for linked data search that includes object co-reference, fact checking, dataset discovery based on connectivity analysis, and connectivity analytics and visualizations. It indexes the entire content of hundreds of datasets in the LOD cloud and recommends relevant datasets by taking into consideration the closure of equivalence relationships based on existing instance (`owl:sameAs`) and class (`owl:EquivalentClass`) equivalence links. As an example, users can request for the K datasets that are most connected with the Hellenic Fire Brigade dataset. A related but slightly different tool is Linklion [30], a semantic web link repository, that is, a catalog of identified links between data sources populated from user-employed link discovery processes, which contains 12.6 million links of 10 different relation types (e.g., `owl:sameAs`, `dbo:spokenIn`) for 449 datasets. The main difference between our work and all the above is that their recommendation processes are based on information about existing links between datasets, while ours is based on the similarity of the spatial distribution of datasets and classes instances. GeoLOD novelties also include on-the-fly class recommendation for spatial datasets in GIS formats and the export of the recommended pair of classes to SILK and LIMES configuration files for direct use in a link-discovery process.

## 3. Design and Methods

GeoLOD consists of two distinct but complementary modules: (a) the Catalog of spatial datasets and classes, and (b) the Recommender of candidate datasets and classes for link discovery. In the following sections, we present the design of and the methods used in each module.

### 3.1. The Catalog

The goal of the Catalog is to provide lists of linked data spatial datasets and classes and methods for their textual and spatially-based retrieval. Each catalog item (a spatial dataset or a class) should be described by its metadata, with an emphasis on describing their spatial characteristics. Users and agents should be able to browse and search the catalog and select an item to view its full description. The main design decisions include (a) the definitions of the terms spatial dataset and spatial class, (b) the identification of the methods for collecting information about available spatial datasets and classes, and (c) the metadata set for describing catalog items.

### 3.1.1. Definitions

An RDF triple is a statement about two resources that follows the `subject predicate object` structure, where `subject` and `object` represent two resources and `predicate` their relation. A set of triples (`S`) is denoted as $S = I \times R \times (I \cup L)$, where `I`, `L`, and `R` represent instances, literals, and relations, respectively, so that `subject` corresponds to an instance, `predicate` to a relation, and `object` to an instance or a literal. With the term spatial dataset, we refer to "a set of RDF triples published, maintained or aggregated provided by a single provider" [32] containing spatial instances, that is, a subject explicitly georeferenced with predicates defined in a spatial vocabulary. A spatial vocabulary defines predicates that allow the representation of an instance location in the form of longitude/latitude coordinates in a well-known Coordinate Reference System (CRS), such as WGS84 (e.g., `Athens hasLongitude ''23.58''`). A spatial class is a subset of a spatial dataset containing spatial instances declared to be instances of a dataset class using the `rdf:type` predicate (e.g., `Athens rdf:type City`). In this work, we search and catalog spatial datasets and their spatial classes, whose instances' locations are expressed as single points, that is, by a longitude and a latitude value, using the W3C Basic Geo [65], GeoVocab [66], GeoSPARQL [67], GeoNames [68], or GeoRSS [69], which are common spatial vocabularies listed in Linked Open Vocabulary (LOV) [70] and LOV4IoT [71]. Furthermore, we search and catalog only those datasets provided by SPARQL endpoints and not by other means, such as RDF dump files. A SPARQL endpoint is an interface that is accessible through a URL and allows access to the triples of a dataset using SPARQL, which is the standard language for querying linked data. Therefore, the terms datasets and SPARQL endpoints are used in the remainder of the paper interchangeably.

### 3.1.2. Data Collection

The initial pool of information about available linked data datasets is formed by parsing the content of other well-known dataset catalogs, namely LOD cloud and DataHub, which provide means for automated consumption of their contents. Specifically, LOD cloud exposes a list of datasets and their metadata at https://lod-cloud.net/lod-data.json (accessed on 16 April 2021) in JSON (an open standard and lightweight data-interchange format), and DataHub allows access to its dataset list using the CKAN API [72] (an API for querying data catalogs). GeoLOD Catalog parses the LOD cloud and DataHub to locate datasets provided through SPARQL endpoints and extract basic metadata, such as their title and endpoint URL. Then, it sends ASK queries to the located SPARQL endpoints to identify which of them uses any of the spatial vocabularies defined in Section 3.1.1. An ASK query is a SPARQL variation that is used to return a true or false answer to the issued query. For example, the ASK query below returns true if the endpoint contains triples that use the `http://www.w3.org/2003/01/geo/wgs84_pos#long` and `http://www.w3.org/2003/01/geo/wgs84_pos#lat` predicates (hereafter, for brevity `geo:long` and `geo:lat`, respectively) of the W3C Basic Geo vocabulary to express the coordinates of an instance (represented by the variable `?subject`).

```
ASK { ?subject <http://www.w3.org/2003/01/geo/wgs84_pos#long> ?x
?subject <http://www.w3.org/2003/01/geo/wgs84_pos#lat> ?y
}
```

After the available spatial datasets have been identified, we retrieve for each dataset its spatial classes by sending SELECT queries to its SPARQL endpoint. A SELECT query is another variation of SPARQL that is used to extract the raw values that answer to the given query. Specifically, we send five SELECT queries (Table 1), one for each vocabulary, to retrieve dataset classes by vocabulary. For example, the W3C Basic Geo SELECT query returns a list of the classes (variable `?class`) that contain instances (variable `?s`) using the `geo:long` and `geo:lat` predicates for expressing their location. We note that if a class uses more than one spatial vocabulary (for example, an instance is georeferenced using W3C Basic Geo and GeoRSS vocabularies), we retrieve the class once in order to avoid

duplicates. Similar SELECT SPARQL queries are sent to calculate the bounding box, the number of spatial instances and other metadata of the spatial classes and datasets, which are presented in the following section.

**Table 1.** SELECT SPARQL queries for retrieving dataset spatial classes.

| Spatial Vocabulary | SELECT Query |
|---|---|
| GeoVocab | `SELECT DISTINCT ?class {`<br>`?geom <http://www.w3.org/2003/01/geo/wgs84_pos#long> ?x.`<br>`?geom <http://www.w3.org/2003/01/geo/wgs84_pos#lat> ?y.`<br>`?s <http://geovocab.org/geometry#geometry> ?geom.`<br>`?s <http://www.w3.org/1999/02/22-rdf-syntax-ns#type> ?class}` |
| GeoSPARQL | `SELECT DISTINCT ?class {`<br>`?s <http://www.opengis.net/ont/geosparql#hasGeometry> ?geom.`<br>`?geom <http://www.opengis.net/ont/geosparql#asWKT> ?wkt.`<br>`?s <http://www.w3.org/1999/02/22-rdf-syntax-ns#type> ?class}` |
| GeoNames | `SELECT DISTINCT ?class {`<br>`?s <http://www.w3.org/2003/01/geo/wgs84_pos#long> ?x.`<br>`?s <http://www.w3.org/2003/01/geo/wgs84_pos#lat>?y.`<br>`?s <http://www.geonames.org/ontology#featureClass> ?class.}` |
| W3C Basic Geo | `SELECT DISTINCT ?class {`<br>`?s <http://www.w3.org/2003/01/geo/wgs84_pos#long> ?x .`<br>`?s <http://www.w3.org/2003/01/geo/wgs84_pos#lat> ?y.`<br>`?s <http://www.w3.org/1999/02/22-rdf-syntax-ns#type> ?class.}` |
| GeoRSS | `SELECT DISTINCT ?class {`<br>`?s <http://www.georss.org/georss/point> ?point.`<br>`?s <http://www.w3.org/1999/02/22-rdf-syntax-ns#type> ?class}` |

### 3.1.3. Item Metadata and GeoVoID

GeoLOD Catalog contains two main categories of items: spatial datasets and spatial classes. Spatial datasets are described by some generic metadata, namely their title, description, SPARQL endpoint URL, and VoID URL (if available), extracted from LOD cloud and DataHub metadata. Moreover, for each dataset, we compute spatial metadata, namely its bounding box, (that is, the minimum bounding rectangle (MBR) that contains all its instance locations), the number of its spatial classes and spatial instances, and the spatial vocabularies found, extracted by sending the appropriate SELECT queries (as described in Section 3.1.2). Spatial classes are described by some generic metadata, namely their URI (Uniform Resource Identifier), label, description, and the dataset that they belong to. For each class, we compute spatial metadata, namely its MBR, the number of its spatial instances, and the spatial vocabulary that it uses. Figure 1 summarizes the metadata set for GeoLOD datasets and classes and their association.

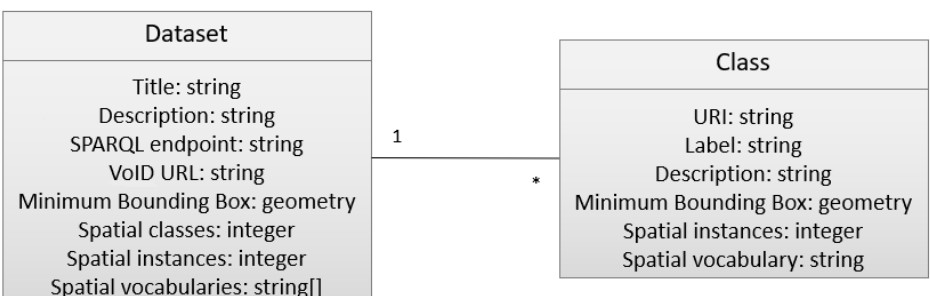

**Figure 1.** GeoLOD Catalog item metadata. A class may belong to 1 dataset and a dataset can contain many (*) classes.

To describe spatial datasets in machine-readable format we designed and introduce GeoVoID, an RDF dataset description vocabulary that extends VoID [32] to express spatial metadata at dataset level. In VoID, a `void:Dataset` class represents the instance of a dataset, which is described by properties, such as `void:entities` (denoting the total number of its entities), `void:classes` (denoting the total number of its classes) and `void:triples` (denoting the total number of its triples). `void:classPartition` is a subset of a `void:Dataset` that contains the description of a certain `rdfs:Class`, which is declared with the property `void:class`. In GeoVoID, each `void:Dataset` class is used to describe a spatial dataset and contains a mandatory `dctetms:spatial` predicate, which denotes the dataset MBR in Well Known Text (WKT) format, which is a markup language for representing vector geometry objects. The newly defined predicates `geovoid:vocabulary`, `geovoid:classes`, and `geovoid:entities` denote the dataset spatial vocabularies, number of spatial classes, and number of spatial instances, respectively, (we remind the reader that VoID corresponding predicates are not restricted to spatial vocabularies, classes, and instances). The `void:classPartition` predicate contains the list of spatial classes of the dataset, where each spatial class is represented by the `void:class` class. Each `void:class` can also contain the `dctetms:spatial`, `geovoid:vocabulary` and `geovoid:entities` predicates to denote the corresponding spatial metadata for a class. The GeoVoID schema is available at http://snf-661343.vm.okeanos.grnet.gr/schemas/geovoid, (accessed on 16 April 2021) and its term definitions are in accordance with the definitions used in this paper; that is, a spatial entity is a georeferenced instance, a spatial class is a class containing one or more spatial instances, and a spatial vocabulary is a vocabulary that can be used for instance georeferencing.

### 3.2. The Recommender

The goal of the GeoLOD Recommender is to provide to each spatial class in the GeoLOD Catalog a ranked list of other spatial classes that may contain related instances, that is, instances that refer to the same real world object or to semantically close objects (e.g., a university and its campus). The recommended pairs of classes can be used as input in a link-discovery process, using tools such as SILK and LIMES, for the establishment of `owl:sameAs` links or other links (e.g., `rdf:seeAlso`) that denote a close semantic relation between instances. Recommender generates recommendation lists for all spatial classes in the background and provides them through the web interface at both class and dataset-level. In addition, it allows the on-the-fly recommendation for datasets that are not listed in the catalog and for non-RDF spatial datasets in well-known spatial data representation formats, such as Shapefile and GeoJSON.

The recommendation process implements the methodology presented in [33], which generates a ranked list of relevant classes for a link discovery process to a given class, based on the similarity of the spatial distribution of their instances. Below, we briefly present the recommendation process, which is analyzed in detail in [33]. Initially, the algorithm builds spatial summaries for each class that capture (a) its spatial extent, by calculating its ConvexHull (the minimum polygon that encloses all instance locations of the class), and (b) the spatial distribution of its instances, by overlaying them on a global pre-computed QuadTree and generating a set of QuadTree cells IDs, that consists of the QuadTree cells IDs that overlap with the instances of the class. QuadTree is a spatial index that segments the world into not-equally-sized cells (each having an ID), where small cells cover areas that present high concentration of linked data instances (such as cities) and large cells cover areas that present low concentration of linked data instances (such as oceans). The algorithm exploits above-described class summaries and computes the similarity of an input (source) class (the class for which someone wants to get recommendations) with each of the other summarized (target) classes. In order to reduce the number of similarity computations, the algorithm filters out target classes that do not spatially overlap with the source class (i.e., their ConvexHulls are disjointed), and their spatial distribution summaries do not have a minimum number of common QuadTree cell IDs (which means that the two

classes share few instances in close proximity). Finally, the algorithm computes a similarity score for the source class and each of the remaining (not filtered out) target classes using one of the similarity metrics proposed in [33]: Number of Common Cells (CC), Jaccard Similarity (JS), Overlap Coefficient (OC), Poisson Distribution Probability (PD), Pointwise Mutual Information (PMI), and Phi Coefficient (PHI). The output of the algorithm is a ranked list of recommended classes to the source class for a link-discovery process. The ranking is determined by the selected metric score so that the higher the similarity between the source and a target class summary sets, the more likely for this pair of classes to contain related instances.

GeoLOD creates summaries and recommendations for all classes in the Catalog by executing the recommendation algorithm described above with the following modifications. Instead of determining the ranking based on one metric, it combines the three most effective metrics, which, according to the evaluation performed in [33], are the Poisson Distribution Probability (PD), the Pointwise Mutual Information (PMI), and the Phi Coefficient (PHI), as follows: the pairs of classes (the source and each of the target classes) are ranked three times based on the similarity score for each metric. Then, the three ranking positions for each pair are summed to compute its combined ranking. For example, if a pair of classes is ranked 1st for the PD, 6th for the PMI, and 3rd for the PHI metric, its combined ranking is 10. Finally, the combined ranking of all pairs is sorted in ascending order to generate the final ranked list of recommended classes.

To further reduce the size of the final lists of recommended classes to a source class, GeoLOD applies an additional filtering condition to exclude pairs of classes that achieve a low similarity score at least for one of the three metrics. The thresholds defined in the following condition were set empirically and are assessed in Section 5.3:

```
PD > 0.90 and PMI > 1 and PHI > 0.02
```

## 4. The GeoLOD Application

### 4.1. Implementation

GeoLOD web interface is available at http://geolod.net/ (accessed on 16 April 2021). The frontend application was developed in *React* [73] and the backend API in *Node.js* [74]. The queries to the SPARQL endpoints were sent with the *Fetch SPARQL endpoint* node.js module [75]. The thumbnails depicting the bounding box of datasets and classes were generated with the *Static Image Mapbox API* [76], and the interactive maps were built on *Leaflet* [77] and *OpenLayers* [78]. The database behind the application is the `PostgreSQL` with the *PostGIS* [79] extension for spatial data management. GeoLOD is hosted in a *Ubuntu 18 LTS 4GB* Virtual Machine, provided by *okeanos*, a GRNET cloud Infrastructure as a Service (IaaS) for Greek academic institutes.

GeoLOD content, that is, the list of spatial datasets and classes with their metadata and the recommendation lists for all classes, is updated automatically every two months, as a background process. For each update, newly identified spatial datasets and classes are imported into the Catalog (according to the methods described in Section 3), and existing datasets and classes are checked for content changes and updated accordingly; for instance, if the number of a class spatial instances has changed, we update its metadata and recalculate its minimum bounding rectangle (MBR).

### 4.2. Use Cases

In the GeoLOD interface (Figure 2), users can browse the complete list of identified spatial datasets and classes or filter them using text and map-based criteria. Upon entering a keyword in the *Filters* dialog box, GeoLOD searches in datasets and classes titles and descriptions, and upon selecting an area in the interactive map, GeoLOD returns datasets and classes whose minimum bounding rectangles intersect or are contained in the selected area, thus allowing users to browse datasets and classes that contain instances in specific geographical areas, such as continents, countries or other user-defined areas. Additionally, users can sort the datasets and classes lists in multiple ways, including sorting by title,

number of instances, and number of recommendations. Upon selecting an item (a dataset or a class), users can view its full description and perform some actions.

On a dataset description page (Figure 3), users can view its title, description, SPARQL endpoint URL, its bounding box on a thumbnail, the spatial vocabularies it uses, the number of spatial entities and classes it contains, the number of recommendations (computed as the sum of recommendations for all dataset classes) and navigate in the list of dataset spatial classes. An icon indicates whether the SPARQL endpoint is currently available (green) or unavailable (red). In addition, users can download its VoID file (if available) and export its GeoVoID description (see Section 3.1.3) and the dataset recommendations list in JSON. The latter can be used for batch link discovery processes and consists of all recommendations for dataset classes. A sample of the JSON file is depicted below: `Recommendations` is the root element, which contains an array of recommendations. Each array object (described inside { and } characters) refers to a recommendation, that is, a pair of classes, and contains the source and the target class SPARQL endpoint (properties `sourceEndpoint` and `targetEndpoint`) and URI (properties `sourceClass` and `targetClass`), respectively.

On a class description page, users can view its label, description, URI, the dataset it belongs to, its bounding box on a thumbnail, the spatial vocabulary it uses, the number of its spatial entities and the list of recommended classes and export the list of recommended classes in JSON. Furthermore, they can download live copies of class instances (extracted on the fly from the SPARQL endpoint) in RDF, JSON, and GeoJSON or browse class spatial instances on an interactive map (Figure 4). We note that the GeoJSON downloads are transformed in order to be readily consumable by a geographic information system (GIS) software, such as QGIS.

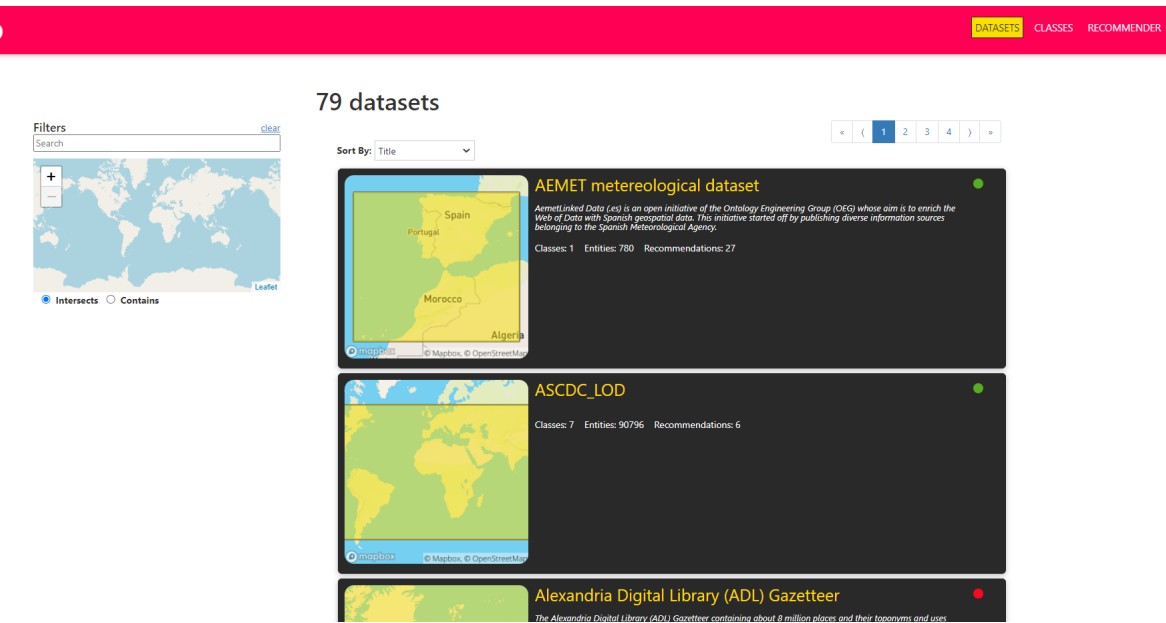

**Figure 2.** GeoLOD home page with the list of linked data spatial datasets.

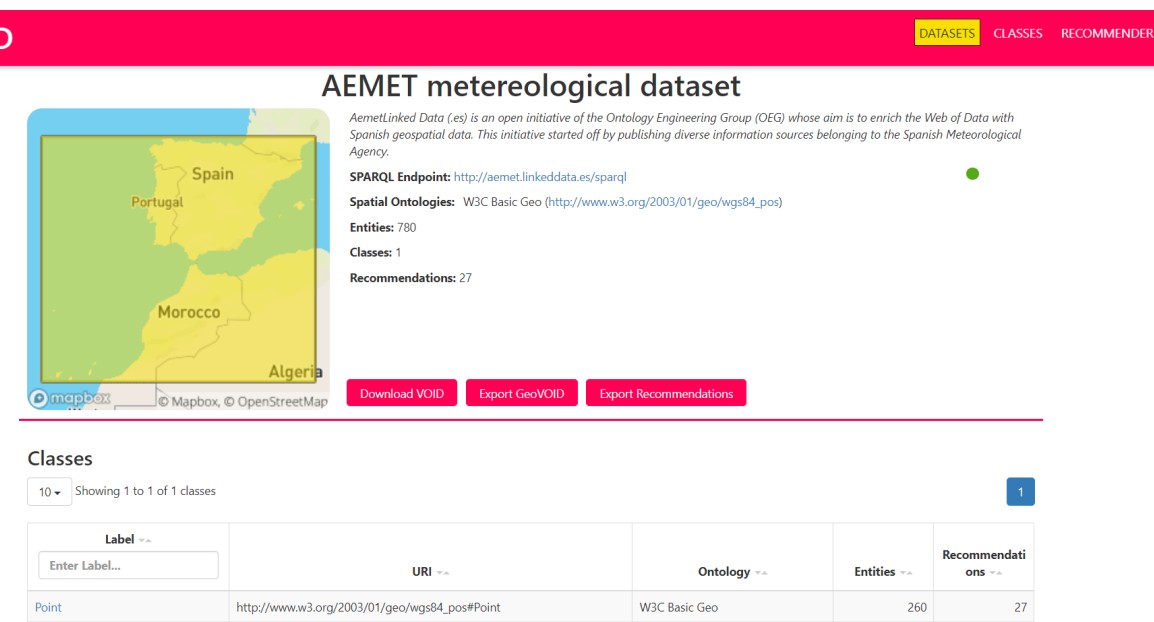

**Figure 3.** The *AEMET* dataset description page.

```
{''Recommendations'':[{
''sourceEndpoint'':''http://aemet.linkeddata.es/sparql'',
''sourceClass'':''http://www.w3.org/2003/01/geo/wgs84_pos#Point'',
''targetEndpoint'':''http://www.linklion.org:8890/sparql'',
''targetClass'':''http://linkedgeodata.org/ontology/AerowayThing''
},{
''sourceEndpoint'':''http://aemet.linkeddata.es/sparql'',
''sourceClass'':''http://www.w3.org/2003/01/geo/wgs84_pos#Point'',
''targetEndpoint'':''http://www.linklion.org:8890/sparql'',
''targetClass'':''http://linkedgeodata.org/ontology/Viewpoint''
},{
...
}]}
```

A snapshot of the recommendation list for a given class (specifically, for the *Point* class of the *AEMET* dataset that contains information about meteorological stations) is depicted in Figure 5. Users can navigate through the list, view details for a recommended class, such as the number of estimated related instances and the ranking order, and export SILK and LIMES configuration files for the pair of classes for direct use in a link discovery process. The configuration files are automatically generated using as input the source (in this example *Point*) and the selected target class SPARQL endpoint URLs and URIs and configured to perform a basic instance matching that (a) "cleans" instance labels, by converting them in lower case and removing special characters, and checks for their *Levenshtein Distance*, which is a typical string similarity metric, and (b) checks the distance of instance locations using the *Euclidean Distance* metric.

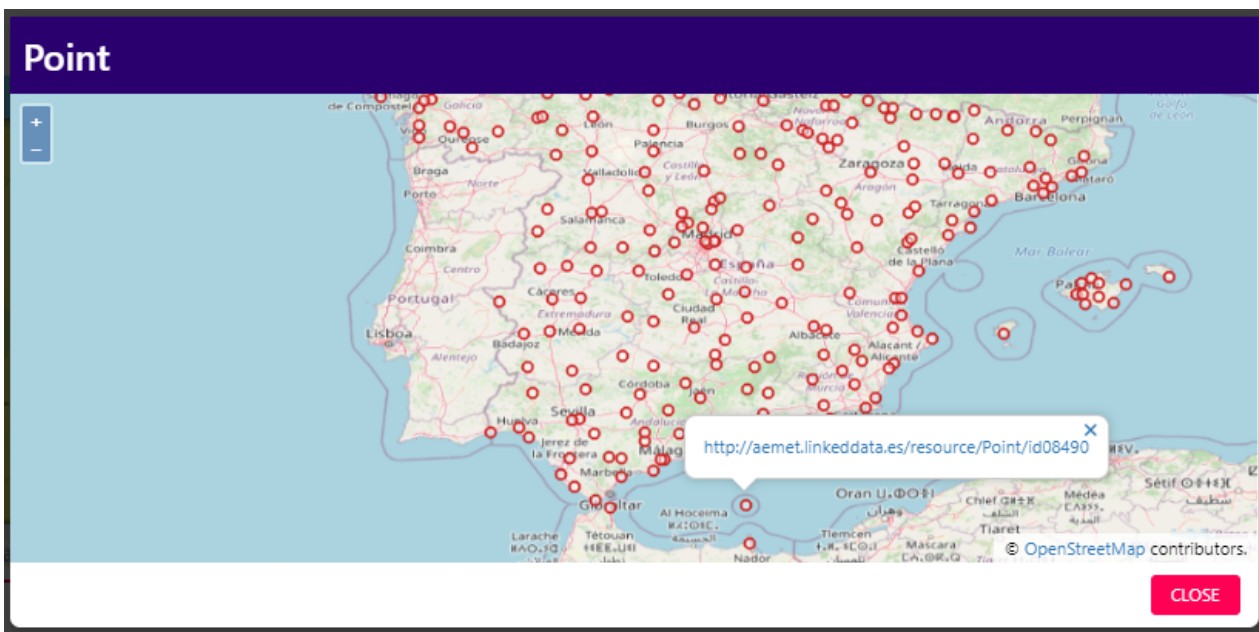

**Figure 4.** *Point* class instances of the *AEMET* dataset on map. The user can click on an instance to get more info in a pop up.

| Label | URI | Dataset | Common Entities | Rank | Link Discovery |
|---|---|---|---|---|---|
| DragLift | http://linkedgeodata.org/ontology/DragLift | LinkLion - A Link Repository for the Web of Data | 3 | 1 | S L |
| Windsock | http://linkedgeodata.org/ontology/Windsock | LinkLion - A Link Repository for the Web of Data | 4 | 2 | S L |
| Windsock | http://linkedgeodata.org/ontology/Windsock | LinkedGeoData | 5 | 3 | S L |
| AirportGate | http://linkedgeodata.org/ontology/AirportGate | LinkLion - A Link Repository for the Web of Data | 4 | 4 | S L |
| AirportGate | http://linkedgeodata.org/ontology/AirportGate | LinkedGeoData | 4 | 5 | S L |
| Beacon | http://linkedgeodata.org/ontology/Beacon | LinkLion - A Link Repository for the Web of Data | 5 | 6 | S L |
| Beacon | http://linkedgeodata.org/ontology/Beacon | LinkedGeoData | 6 | 7 | S L |
| ParkingSpace | http://linkedgeodata.org/ontology/ParkingSpace | LinkLion - A Link Repository for the Web of Data | 3 | 8 | S L |
| terms#Stop | http://vocab.gtfs.org/terms#Stop | Aragon Interoperable Information Structure EI2A - aragon open data | 12 | 9 | S L |
| Lighthouse | http://linkedgeodata.org/ontology/Lighthouse | LinkLion - A Link Repository for the Web of Data | 13 | 10 | S L |

**Figure 5.** The ranked class recommendations list for the *Point* class of the *AEMET* dataset.

Figure 6 shows the on-the-fly recommender user interface for generating recommendations for datasets that are not listed in the GeoLOD Catalog. Initially, users select the type of the input dataset that can be a SPARQL endpoint, a GeoJSON, or a Shapefile (step 1). For the first case, they enter the URL of the endpoint and select a class from the automatically populated list; for the other cases, they upload the corresponding files. GeoLOD parses the input dataset, builds in real time the required summaries and metadata and generates a preview (step 2). Finally, users click the *Get Recommendations* button and GeoLOD searches in the Catalog to return the list of recommended classes for link discovery.

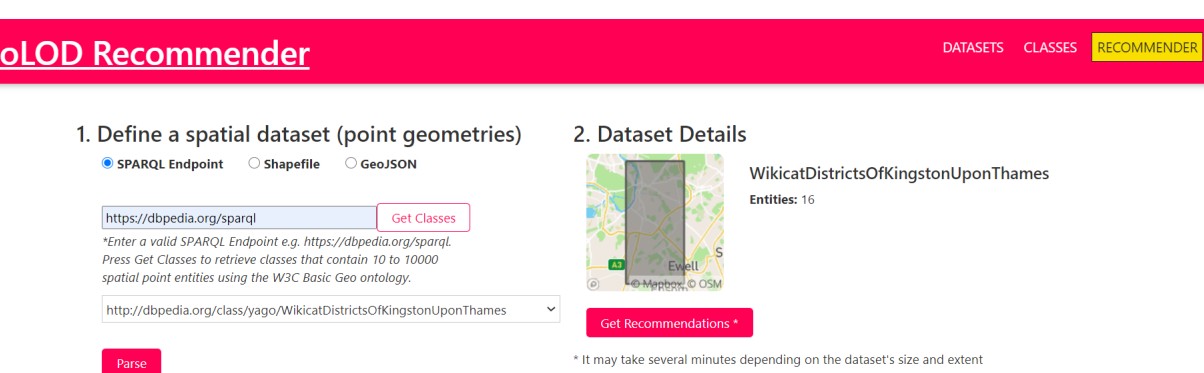

**Figure 6.** The on-the-fly recommender interface.

*4.3. REST API*

GeoLOD provides a REST API that can be used by software agents. Table 2 lists the names, the request URI (the left part of the URI is http://snf-661343.vm.okeanos.grnet.gr accessed on 16 April 2021), and the descriptions of the main services.

**Table 2.** GeoLOD REST services.

| Service Name | Request URI | Description |
| --- | --- | --- |
| GeoLOD Description | /api/download/dcat | Returns a DCAT-compliant turtle file that contains general information about GeoLOD and the list of the datasets in the Catalog |
| Dataset List | /api/datasets | Returns, in JSON, the list of datasets with their metadata (including internal dataset IDs) in the GeoLOD Catalog |
| Dataset Description | /api/datasets/<ID> | Returns, in JSON, the specified dataset metadata with the list of its classes. The dataset ID is a variable corresponding to the internal dataset ID. (e.g., http://snf-661343.vm.okeanos.grnet.gr/api/datasets/915 accessed on 16 April 2021, returns the metadata for the *AEMET* dataset) |
| Class List | /api/classes | Returns, in JSON, the list of classes with their metadata (including internal classes IDs) in the GeoLOD Catalog. |
| Class Description | /api/classes/<ID> | Returns, in JSON, the specified class metadata with the list of its recommended classes. The class ID is a variable corresponding to the internal class ID. (e.g., http://snf-661343.vm.okeanos.grnet.gr/api/classes/139090 accessed on 16 April 2021, returns the metadata for the *CaveEntrance* class of *Linklion* dataset). |
| Dataset GeoVoID | /api/download/geovoid/<ID> | Returns, in turtle format, the GeoVoID description of the specified dataset. |
| Dataset Recommendations | /api/download/ datasetrecommendations/<ID> | Returns, in JSON, the list of recommendations for all specified dataset classes. |
| Class Recommenations | api/download/ classesrecommendations/<ID> | Returns, in JSON, the list of recommendations for the specified class. |

## 5. Results

In this Section, we present statistics that provide insights into the characteristics of spatial datasets in the Web of Data (Section 5.1) and the potential interlinkings between spatial datasets and classes based on GeoLOD recommendations (Section 5.2). In Section 5.3, we

assess the applicability of the Recommender by examining the relation between GeoLOD class recommendations and LIMES instance recommendation for different algorithm variations. Finally, we present the findings of the system usability study that we performed to evaluate GeoLOD application (Section 5.4).

### 5.1. Catalog Statistics

In November 2020, LOD cloud and DataHub contained 478 and 723 datasets provided through SPARQL endpoints, respectively. Many datasets are listed in both catalogs, and some are provided through the same endpoint. GeoLOD identified 629 unique SPARQL endpoints from both catalogs. After sending simple SPARQL ASK queries to each (see Section 3.1.2), 477 returned an error response, such as URL unavailable or timeout, indicating that approximately only 24% of the total SPARQL endpoints found in LOD cloud and DataHub are active. Of the remaining 152 active endpoints, 60 responded true; that is, they contain a spatial vocabulary, which means that approximately 39% of the active endpoints contain georeferenced information.

In the following pages, we analyze the content of the identified spatial datasets, and we present statistics that reveal the availability and distribution of the spatial information in the Web of Data. Initially, we sent SPARQL SELECT queries to the 60 SPARQL endpoints in order to retrieve their spatial classes and collect statistics, namely, the number of its total classes, spatial classes, total instances, and spatial instances. During the investigation, we found endpoints that could not respond to the issued SELECT queries and endpoints that are duplicates or mirror other endpoints, and we excluded them from subsequent analysis. We also removed classes that contain very few instances (less than 5), because these classes cannot be used for generating recommendations, or too many instances (more than 100,000) in order to avoid high computational costs. Finally, we excluded the DBpedia dataset from our analysis, which contains 22,742 spatial classes (approximately seven times more than the sum of spatial classes of the other datasets) and more than 1 million spatial instances.

Due to the above restrictions, we finally analyzed 40 SPARQL endpoints, presented in Table 3. The total number of identified spatial classes is 3418, that is, approximately 5% of the total classes (66,571) provided by the 40 identified spatial datasets. Accordingly, we identified approximately 77 million georeferenced instances, that is, approximately 18% of the total instances (424 million) provided by the same datasets. Table 3 reveals that the biggest providers of spatial information are the *LinkedGeoData* and *Linklion* datasets, containing 952 and 902 spatial classes and more than 48 and 20 million spatial instances, respectively.

Next, we present information about the spatial characteristics of linked data datasets and classes. Table 4 presents the statistical distribution of datasets and classes by the size of their spatial extents (i.e., their mininum bounding rectangles), classified into five categories, each representing an area roughly equal to a common geographical notion, ranging from small areas, covering medium sized cities, to large areas, covering the whole world. Most datasets and classes are "global" or cover areas approximately equal to continents (about 78% of datasets and 87% of classes), which shows that most linked data providers publish large area datasets and that few providers published local datasets. Furthermore, by inspecting classes content on the GeoLOD interactive map, we noticed that in many cases, the population completeness, that is, the percentage of all real-world objects of a particular type that are represented in a class [80], regarding spatial instances at local level is small. The implication of these findings is that local mapping organizations have not yet adopted linked data technologies. Figure 7 shows the spatial extents of all spatial datasets and their density all over the world and indicates that most non-global-scale datasets are located in and around Europe. A closer examination of Figure 7 reveals potential georeferencing errors for some datasets. For example, there is a dataset that extends in a small area around zero longitude and latitude in the Gulf of Guinea at the Atlantic Ocean and another whose MBR is a thin line that starts in the Pacific Ocean, east of South America, and ends in Australia.

**Table 3.** Number of total and spatial classes, total and spatial instances for 40 SPARQL endpoints. N/A denotes that the number could not be retrieved because of errors returned from the endpoint.

| SN | DATASET | CLASSES | | INSTANCES | |
|----|---------|---------|---------|-----------|---------|
| | | **TOTAL** | **SPATIAL** | **TOTAL** | **SPATIAL** |
| 1 | AEMET metereological dataset | 35 | 1 | N/A | 260 |
| 2 | AragoDBPedia - aragon open data | 164 | 1 | N/A | 357,678 |
| 3 | Datos.bcn.cl | 500 | 6 | 5,303,750 | 830 |
| 4 | DBpedia in Basque | 223 | 20 | 1,168,342 | 51,547 |
| 5 | DBpedia in Dutch | 666 | 142 | 6,718,584 | 252,310 |
| 6 | DBpedia in French | 442 | 188 | 6,015,375 | 225,030 |
| 7 | DBpedia in German | 557 | 123 | 6,682,441 | N/A |
| 8 | DBpedia in Greek | 14,439 | 245 | 2,852,513 | 12,609 |
| 9 | DBpedia in Japanese | 727 | 100 | 4,254,851 | 36,827 |
| 10 | DBpedia in Spanish | 748 | 126 | 5,249,003 | 36,800 |
| 11 | Dutch Ships and Sailors | 92 | 11 | N/A | 42,810 |
| 12 | El Viajero's tourism dataset | 67 | 1 | 1,019,390 | 643 |
| 13 | Environment Agency Bathing Water Quality | 93 | 7 | 801,310 | 1216 |
| 14 | European Nature Information System | 629 | 13 | N/A | 1,129,574 |
| 15 | European Pollutant Release and Transfer Register | 375 | 10 | 78,719,353 | 3,325,006 |
| 16 | EuroVoc | 413 | 1 | 91,726,256 | 672 |
| 17 | Geological Survey of Austria (GBA)—Thesaurus | 23 | 1 | 3004 | 130 |
| 18 | Indicators Academic Process 2017 | 79 | 7 | 516,097 | 159 |
| 19 | Isidore | 62 | 4 | 20,662,124 | 4101 |
| 20 | ISPRA—The administrative divisions of Italy | 99 | 4 | 449,218 | 23,211 |
| 21 | Linked Logainm | 114 | 38 | 214,423 | 108,065 |
| 22 | LinkedGeoData | 1908 | 952 | N/A | 48,249,489 |
| 23 | LinkLion | 1137 | 902 | 138,806,633 | 20,006,546 |
| 24 | Lotico | 23 | 7 | N/A | N/A |
| 25 | MONDIS | 662 | 3 | 12,855 | 10 |
| 26 | MORElab | 223 | 20 | 1,168,342 | 51,547 |
| 27 | Open Data Communities—Lower layer Super Output Areas | 334 | 14 | 7,912,454 | 2,694,723 |
| 28 | OpenMobileNetwork | 156 | 1 | 934,551 | 357,298 |
| 29 | OxPoints (University of Oxford) | 106 | 5 | 114,813 | 1457 |
| 30 | Serendipity | 607 | 109 | N/A | 61,845 |
| 31 | Shoah victims? names | 200 | 35 | 1,956,021 | 13,974 |
| 32 | Social Semantic Web Thesaurus | 521 | 16 | 14,214 | 54 |
| 33 | Spanish Linguistic Datasets | 57 | 1 | 2,977,659 | 764 |
| 34 | Suface Forestire Mondiale 1990–2016 | 4188 | 2 | 187,608 | 100 |
| 35 | TAXREF-LD: Linked Data French Taxonomic Register | 1921 | 10 | 8,255,730 | 996 |
| 36 | Test Site, LOD Lab 317 | 98 | 4 | 5,669,728 | 115,225 |
| 37 | URIBurner | 33,656 | 262 | 22,175,094 | 456,360 |
| 38 | Verrijkt Koninkrijk | 52 | 12 | 329,621 | 42,831 |
| 39 | WarSampo | 90 | 10 | 1,797,432 | 33,685 |
| 40 | World War 1 as Linked Open Data | 85 | 4 | 14,644 | 883 |
| | SUM | 66,571 | 3418 | 424,674,433 | 77,697,265 |

**Table 4.** Datasets and classes classified by the size of their spatial extent.

| Spatial Extent (Km$^2$) | Datasets | Classes |
|-------------------------|----------|---------|
| City-level (<1 K) | 1 | 54 |
| Region level (1 K–100 K) | 3 | 218 |
| Country level (100 K–1000 K) | 5 | 183 |
| Continent level (1000 K–50,000 K) | 17 | 1316 |
| World level (>50,000 K) | 14 | 1647 |

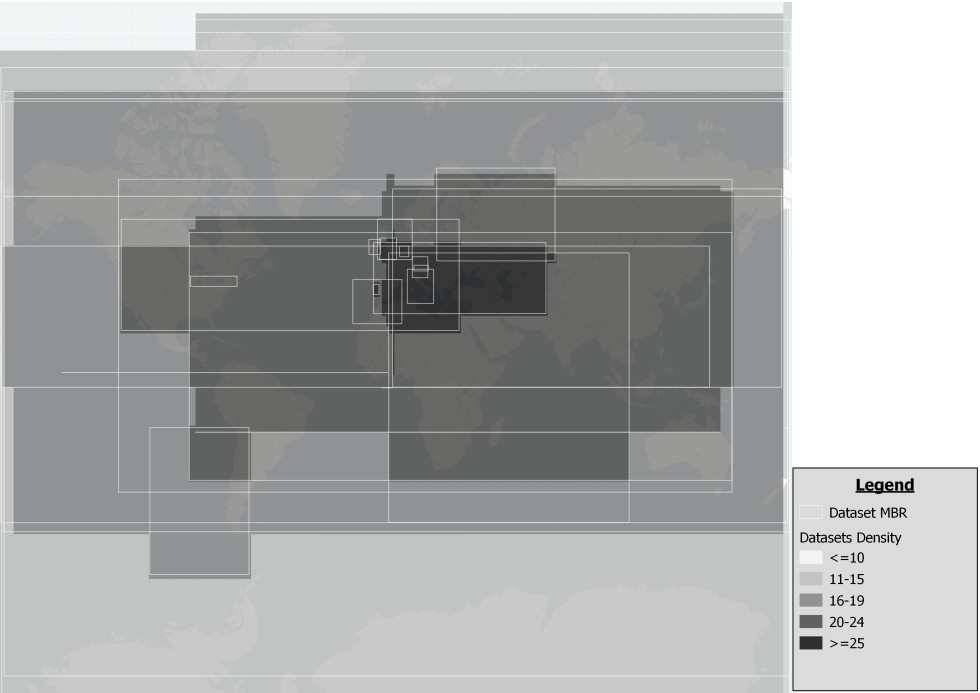

**Figure 7.** Spatial datasets minimum bounding rectangles and density.

We close this section by presenting two more findings. Regarding the use of spatial vocabularies, the most used spatial vocabulary is the W3C Basic Geo, which is used in all datasets (40) that were examined and in 3345 classes. Ten datasets also use the Geonames and one dataset the GeoVocab vocabularies in 36 and 37 spatial classes, respectively. GeoRSS is used with W3C Basic Geo in 15 datasets, and no dataset was found that uses the GeoSPARQL vocabulary. Concerning the availability of VoID files, of the 629 identified datasets in LOD cloud and DataHub provided through SPARQL endpoints, only 236 were found to publish a VoID description, and, of the 40 datasets listed in Table 3, the respective number is 11, which shows that providers usually do not provide VoID description of their datasets. Furthermore, in none of the provided VoID descriptions did we find information for describing the spatial aspects that we present in this paper, such as dataset bounding boxes.

*5.2. Recommender Statistics*

In this section, we analyze the outcome of the GeoLOD Recommender that provides insights into the potential interlinking of linked data spatial datasets and classes. In particular, we executed the recommendation algorithm for 3418 spatial classes provided by the 40 spatial datasets (Table 3) using the ranking mechanism and filtering condition presented in Section 3.2.

Table 5 presents the results of the recommendation algorithm summarized by dataset. For each dataset, it shows (a) the number of its spatial classes as listed in Table 3 (column DC), (b) the number of dataset classes for which there are recommendations (column DCR), (c) the number of recommendations to other dataset classes (column OCR), and (d) the number of recommendations to other datasets (column ODR). It is worth noting that the numbers in Table 5 refer to GeoLOD recommendations (with the specific algorithm parameters) and not to the correctly recommended classes and datasets. Furthermore, the presented statistics include only recommendations for other dataset classes and not for classes provided by the same dataset as the source class. Finally, we note that columns DCR, OCR, and ODR can be read in two ways; the number of dataset classes for which there are recommendations (column DCR) denotes the number of dataset classes for which there are recommendations to classes of other datasets (outbound recommendations) but

also the number of dataset classes for which there are recommendations from classes of other datasets (inbound recommendations).

**Table 5.** GeoLOD Recommendations statistics (DC = Number of dataset classes, DCR = Number of dataset classes for which there are recommendations, OCR = Number of recommendations to other dataset classes, ODR = Number of recommendations to other datasets).

| SN | DATASET | DC | DCR | OCR | ODR |
|----|---------|----|-----|-----|-----|
| 1 | AEMET metereological dataset | 1 | 1 | 26 | 5 |
| 2 | AragoDBPedia—aragon open data | 1 | 1 | 31 | 7 |
| 3 | Datos.bcn.cl | 6 | 6 | 777 | 7 |
| 4 | DBpedia in Basque | 20 | 20 | 1380 | 19 |
| 5 | DBpedia in Dutch | 142 | 102 | 1620 | 19 |
| 6 | DBpedia in French | 188 | 144 | 4569 | 26 |
| 7 | DBpedia in German | 123 | 98 | 1881 | 22 |
| 8 | DBpedia in Greek | 245 | 174 | 1864 | 25 |
| 9 | DBpedia in Japanese | 100 | 92 | 2407 | 15 |
| 10 | DBpedia in Spanish | 126 | 116 | 1857 | 11 |
| 11 | Dutch Ships and Sailors | 11 | 11 | 349 | 15 |
| 12 | El Viajero's tourism dataset | 1 | 1 | 135 | 11 |
| 13 | Environment Agency Bathing Water Quality | 7 | 7 | 125 | 4 |
| 14 | European Nature Information System | 13 | 13 | 628 | 23 |
| 15 | European Pollutant Release and Transfer Register | 10 | 10 | 891 | 26 |
| 16 | EuroVoc | 1 | 1 | 89 | 10 |
| 17 | Geological Survey of Austria (GBA)—Thesaurus | 1 | 1 | 38 | 4 |
| 18 | Indicators Academic Process 2017 | 7 | 1 | 2 | 2 |
| 19 | Isidore | 4 | 4 | 146 | 13 |
| 20 | ISPRA—The administrative divisions of Italy | 4 | 4 | 200 | 10 |
| 21 | Linked Logainm | 38 | 31 | 573 | 13 |
| 22 | LinkedGeoData | 952 | 900 | 28,603 | 34 |
| 23 | LinkLion | 902 | 885 | 28,610 | 37 |
| 24 | Lotico | 7 | 7 | 455 | 25 |
| 25 | MONDIS | 3 | 3 | 18 | 2 |
| 26 | MORElab | 20 | 20 | 1380 | 19 |
| 27 | Open Data Communities—Lower layer Super Output Areas | 14 | 14 | 362 | 7 |
| 28 | OpenMobileNetwork | 1 | 1 | 117 | 8 |
| 29 | OxPoints (University of Oxford) | 5 | 5 | 48 | 5 |
| 30 | Serendipity | 109 | 107 | 2889 | 19 |
| 31 | Shoah victims? names | 35 | 20 | 643 | 13 |
| 32 | Social Semantic Web Thesaurus | 16 | 10 | 59 | 5 |
| 33 | Spanish Linguistic Datasets | 1 | 1 | 10 | 3 |
| 34 | Suface Forestire Mondiale 1990–2016 | 2 | 0 | 0 | 0 |
| 35 | TAXREF-LD: Linked Data French Taxonomic Register | 10 | 4 | 30 | 3 |
| 36 | Test Site, LOD Lab 317 | 4 | 4 | 58 | 4 |
| 37 | URIBurner | 262 | 186 | 3052 | 22 |
| 38 | Verrijkt Koninkrijk | 12 | 11 | 373 | 15 |
| 39 | WarSampo | 10 | 10 | 621 | 10 |
| 40 | World War 1 as Linked Open Data | 4 | 3 | 82 | 9 |
| | SUM | 3418 | 3029 | 86,998 | 530 |
| | AVERAGE | 85.45 | 75.73 | 2175.00 | 13.25 |

GeoLOD recommends one or more relevant classes for link discovery for 3029 classes, that is, for approximately 89% of all classes. This means that GeoLOD does not find recommendations for only 389 (out of 3418) classes. The 3029 classes belong to 39 different datasets, which means that for all datasets (except *Suface Forestière Mondiale 1990–2016*) GeoLOD produces recommendations. The total number of class recommendations is 86,998

(we note that class recommendations including same dataset classes is 164,782), and thus, the average classes recommendations per class is 25.45, which means that each class gets recommendations for (or from) approximately 0.75% of the total linked data classes (25.45 of 3418). At dataset level, each dataset has on average 2175 recommendations to classes of other datasets and 13.25 recommendations to other datasets, which means that each dataset gets recommendations to (or from) 13.25 other datasets, that is, approximately 33% of the total identified spatial datasets. Table 5 shows that *LinkedGeoData* and *Linklion* are hub datasets, regarding the number of recommendations they have to (or from) other datasets, having recommendations to 34 and 37 other datasets, respectively.

Regarding the execution time of the recommendation algorithm, it requires approximately one day to build summaries and 44 days to generate the recommendation lists for the 3418 classes. Thus, it requires on average 18 min to generate the recommendation list for each class, although the execution time depends on the source class size and spatial extent, ranging from a few seconds to several minutes. We note that this is also the average execution time of the GeoLOD on-the-fly recommender, which builds summaries and generates reccomendations in real time.

*5.3. Recommender Applicability Assessment*

In [33], we evaluated the effectiveness of the recommendation methodology that is implemented in GeoLOD, and we showed that the three most effective metrics are PD (Poisson Distribution Probability), PMI (Pointwise Mutual Information), and PHI (Phi Coefficient) and that the most effective, PD, generates ranked lists of recommended classes with 62% mean average precision, approximately 35% higher than simple baselines. In this work, we assess the benefits of employing GeoLOD Recommender as a preparatory step in link-discovery processes regarding its applicability and gains in time and we examine the effect of the ranking mechanism and the filtering condition that we presented in Section 3.2. For this reason, we execute three recommendation algorithm variations and estimate the percentage of GeoLOD recommended pairs of classes for which the LIMES link discovery framework finds possible instance links. We recall that LIMES recommends possible links between instances of two instance sets (in this case, classes), whereas GeoLOD recommends possible pairs of classes for which instance links can be recommended. Therefore, the higher the number of GeoLOD recommended pairs of classes for which LIMES recommends instance links, the higher the quality and usefulness of GeoLOD recommendations.

We execute the first (default) recommender algorithm variation as follows. We initially selected, from the list of recommendations presented in Section 5.2, a random sample of 5000 (out of the total 86,998) recommendations, that is, pairs of classes. To simplify the configuration of LIMES, we restricted on classes using the W3C Basic Geo spatial vocabulary. We then imported the sample set of recommendations as a batch process to LIMES, each configured with the corresponding source and target endpoint URL and class URI and with the following matching rule:

```
AND(levenshtein(a.rdfs:label,b.rdfs:label)|0.8, euclidean(a.slat|slong,b.tlat|tlong)|0.8)
```

that recommends a link between two instances when the *(Normalized) Levenshtein Distance* of the instances labels is greater than 0.8 and the LIMES euclidean metric of the instances location is greater than 0.8, which corresponds to a euclidean distance of 0.25 degrees, equal to 25 km at the equator in the WGS84 Coordinate Reference System. We should note that the labels' distance is measured after "cleaning" them, that is, converting them into lower case and removing special characters using the LIMES *regularAlphabet* function.

We examined two more aspects of the GeoLOD recommendations, namely, (a) the quality of Top-1 GeoLOD recommendations by importing in LIMES only the top ranked recommendations for each class and (b) the effect of the final filtering condition of the recommendation algorithm by importing in LIMES only those recommendations that satisfy the following (more strict compared to the default) condition:

```
PD>0.95 and PMI>3 and PHI>0.2
```

As baseline, we input in LIMES 5000 pairs of classes randomly selected from the GeoLOD Catalog. Since these pairs are not necessarily GeoLOD recommendations, we compare the applicability of the GeoLOD recommendations against random pairs of classes. Table 6 summarizes the experimental results for the three GeoLOD recommendation algorithm configurations and the baseline. For each, it shows the number of pairs of classes that were used as input in LIMES (column 2, LIMES executions), the number of pairs of classes for which LIMES found one or more possible instance links (that we call them hits) and its percentage to the number of LIMES executions (columns 3 and 4), and the average number of LIMES instance links recommendations per hit (column 5).

**Table 6.** GeoLOD recommender evaluation using LIMES.

|  | (2) LIMES Executions | (3) Hits | (4) Hits (%) | (5) Average Instance Links per Hit |
|---|---|---|---|---|
| (Default) GeoLOD recommendations | 5000 | 2799 | 55.98% | 4003 |
| Top-1 GeoLOD recommendations | 2799 | 1947 | 69.56% | 9339 |
| Strict GeoLOD recommendations | 3858 | 2650 | 68.68% | 13,119 |
| Random Pairs of Classes (Baseline) | 5000 | 344 | 6.88% | 303 |

The percentage of pairs of classes for which LIMES recommends instance links for the GeoLOD class recommendations (column 4), regardless of configuration, outperforms the respective percentage of the randomly generated pairs of classes (baseline). Particularly, 55.98% of the default, 69.56% of the Top-1, and 68.68% of the strict GeoLOD recommendations contain link recommendations according to LIMES basic link specification. Strict GeoLOD recommendations present a higher percentage of hits compared to the default GeoLOD recommendations, but the recommendation list is significantly reduced (3858 recommendations compared to 86,998), which means that default GeoLOD recommendations include more false positives but, at the same time, more true positives compared to the strict GeoLOD recommendations. In the GeoLOD frontend, we use the default recommendation algorithm condition (`PD > 0.90 and PMI > 1 and PHI > 0.02`) because the recommendations are ranked and users can decide how far they want to go in the recommendation lists to find all the recommended pairs of classes for which LIMES recommends instance links. However, with minor modifications to the GeoLOD fronted, users could select between a strict or loose filtering condition.

We should note that if, for a pair of classes, LIMES recommends one or more instances' links, this does not necessarily mean that this pair of classes indeed contain related instances. Conducting rigorous experiments to evaluate the quality of LIMES recommendations, that is, whether instance link recommendations truly correspond to related instances, is out of the scope of this paper. Nevertheless, we can assume that if a pair of classes contains many LIMES instance link recommendations, it is more possible to truly contain related instances than a pair of classes with few LIMES instance link recommendations. Based on the above assumption, we compare the GeoLOD recommendation algorithm variations by examining the average number of instances links recommendations per relevant pair of classes. Table 6 shows that for random pairs of spatial classes the average number of LIMES instances links recommendations per pair (column 5) is 303, while for GeoLOD recommended pairs, the respective number is much higher for all GeoLOD recommendations configurations. Specifically, the highest average is achieved by the strict variation, presenting 13,119 instance links recommendations per pair of recommended classes. Therefore, we can conclude that GeoLOD (especially, the strict variation) is more likely to recommend pairs of classes that truly contain related instances than the random baseline.

Finally, we discuss the search space reduction of the GeoLOD Recommender and the time saved when it is used as a preparatory step of a link-discovery process. The number of pairwise class comparisons needed for finding all possible instance links for all

identified spatial classes is 3418 × 3418 = 11,682,724. GeoLOD generates approximately 165,360 recommendations (including classes from same datasets), and thus reduces the search space approximately 70 times. In our experiments, LIMES required approximately one hour to compare 1000 pairs of classes for instance link recommendations, and thus, to compare all possible pairs of classes in GeoLOD Catalog, LIMES requires 486 days (with 6.88% probability of finding a pair of classes with links), while comparing the GeoLOD recommended pairs requires 7 days (with 55.98% probability of finding a pair of classes with links). For a single class, the execution time for instance link discovery is approximately 3.5 h (for examining 3418 pairs of classes), while, using the on-the-fly GeoLOD Recommender, it is 18 min (the average time GeoLOD requires to generate recommendations for a single class) plus, on average, 3 min (for the 50 recommended pairs of classes, the average GeoLOD recommendations per class including same dataset recommendations, comparisons in LIMES), that is, approximately 21 min.

### 5.4. Usability Study

As already stated, GeoLOD user interface mainly targets linked data experts and GIS professionals in order to facilitate them during their linked data exploration and link-discovery processes. For this reason, we conducted a system usability study to assess how each category of users perceives GeoLOD and to identify strong and weak features in order to improve the application. The study is based on the System Usability Scale (SUS) [81], which consists of 10 questions to be rated on a five-point scale ranging from strongly disagree to strongly agree, among which five are positive statements and the remaining are negative. An adjective rating was added as an eleventh question to collect user ratings of the perceived usability according to a seven-point scale with different wordings [82]. The participants were selected to be experts in either linked data, GIS, or both domains. Initially, invitations were sent to academia and business people with known experience in these domains, and those who responded positively participated on a voluntary basis. The study was completed in two web sessions, held on different days, allowing the participants to choose based on their availability. At the beginning of each session, we explained the purpose of the study and briefly introduced the GeoLOD application. Then, participants had some time to get familiar with the application and to execute some indicative tasks, such as:

- Search for datasets that contain data in a specific geographic area (e.g., Spain);
- View the description and the list of classes of a dataset of their choice;
- View the description and the list of recommendations of a class of their choice;
- View the instances of their selected class on the map;
- Get recommendations for a uncatalogued endpoint, for example `https://dbpedia.org/sparql` (only for linked data experts);
- Get recommendations for a shapefile that they own (only for GIS experts).

Finally, participants completed the online SUS with an adjective rating questionnaire. Each session concluded with a short discussion, where participants expressed their general comments and proposals for the improvement of the application.

In the study, in total, 41 users participated; 11 users perceived themselves as linked data experts and 30 as GIS experts. Of the 41 users, only four declared that they are experts on both domains. Table 7 summarizes the results of the usability study. The first two rows show the results for each category of users and the last row contains the total results. The mean SUS score indicates the overall level of usability, where the minimum possible score is 0 and the maximum possible is 100. The mean SUS score for all participants is 68.48, and the respective score for linked data experts is higher (81.36) than for GIS experts (63.75). The adjective rating corresponds to the results of the 11th seven-scale question "Overall, I would rate the user-friendliness of this product as:", where 1 means `Worst Imaginable` and 7 `Best Imaginable`. The mean adjective rating for all participants is 5.17, and the respective rating for linked data experts is also higher (5.64) than of GIS experts (5.00).

Table 8 and Figure 8 present with more analysis the results of the SUS and the adjective rating questionnaire per question and user category.

**Table 7.** Standard Usability Scale (SUS) with adjective rating questionnaire results.

| Focus Group | Participants | Min SUS | Max SUS | Mean SUS | Mean Adj. Rating (1–7) |
|---|---|---|---|---|---|
| Linked Data experts | 11 | 55 | 100 | 81.36 | 5.64 |
| GIS experts | 30 | 40 | 97.5 | 63.75 | 5.00 |
| All | 41 | 40 | 100 | 68.48 | 5.17 |

**Table 8.** Standard Usability Scale (SUS) questionnaire results per question in the scale 1 (Strongly disagree) to 5 (Strongly agree).

| Question | Linked Data Experts | GIS Experts | All |
|---|---|---|---|
| 1. I think that I would like to use this system frequently. | 3.82 | 3.03 | 3.24 |
| 2. I found the system unnecessarily complex. | 1.64 | 2.07 | 1.95 |
| 3. I thought the system was easy to use. | 4.27 | 3.33 | 3.59 |
| 4. I think that I would need the support of a technical person to be able to use this system. | 2.09 | 2.30 | 2.24 |
| 5. I found the various functions in the system were well integrated. | 4.18 | 3.77 | 3.88 |
| 6. I thought there was too much inconsistency in this system. | 1.45 | 2.10 | 1.93 |
| 7. I imagine that most people would learn to use this system very quickly. | 4.55 | 3.33 | 3.66 |
| 8. I found the system very awkward to use. | 1.82 | 1.83 | 1.83 |
| 9. I felt very confident using the system. | 4.18 | 3.30 | 3.54 |
| 10. I needed to learn a lot of things before I could get going with this system. | 1.45 | 2.97 | 2.56 |

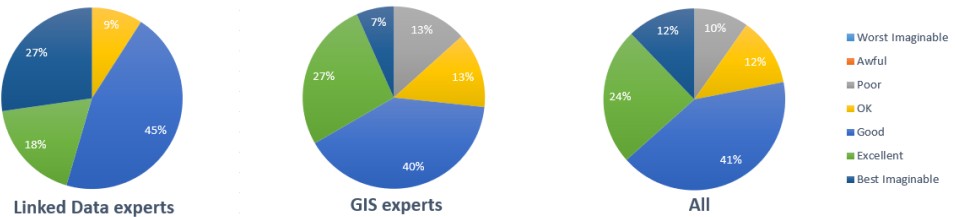

**Figure 8.** Adjective ratings per user category: Linked data experts (**left**), GIS experts (**center**), all (**right**).

The results of the study indicate that the opinion of the users regarding GeoLOD usability and friendliness is good and almost excellent among linked data experts. Furthermore, the responses to the first question of the SUS questionnaire shows that they believe that the application is useful. During the discussion, it emerged that users, especially those who were not linked data experts, would like more guidance (e.g., by including tooltips or explanatory text in the user interface), since they are not familiar with some terms, such as VoID and SPARQL endpoint. Some other proposals included the improvement of the

on-the-fly recommender response times, responsiveness for mobile devices, and inclusion of datasets that contain polygon geometries.

## 6. Discussion and Conclusions

In this paper, we presented GeoLOD, a web catalog of spatial linked data datasets and classes and a recommender for datasets and classes that may contain related spatial instances. GeoLOD addresses user needs for linked data search, taking into account the spatial characteristics of datasets, and is the first exhaustive catalog and recommender exclusively for spatial datasets and classes. It provides a user-friendly interface and an API for automated content consumption. It currently contains metadata for 79 spatial datasets and 5130 spatial classes, identified by parsing the LOD cloud and DataHub catalogs. It also provides more than 166,000 recommendations for pairs of classes that may contain the same or closely related instances and an on-the-fly recommender for user-submitted SPARQL endpoints and spatial datasets in GeoJSON and Shapefile formats. The catalog and the recommendations lists are updated in the background every two months.

GeoLOD is compliant with the linked data standards for describing catalogs and datasets, providing its content in DCAT and datasets descriptions in GeoVoID. GeoVoID was introduced in this paper and extends VoID to describe spatial characteristics of datasets. In the results section, we have presented statistics about the availability of SPARQL endpoints and VoID descriptions that confirm other recent studies [25,26,51,83]; few datasets are accompanied by their VoID descriptions, and furthermore, there is no description of their spatial characteristics, such as their bounding box or the number of their georeferenced instances. GeoLOD fills this gap by automatically generating GeoVoID descriptions for each dataset in the Catalog. Our analysis reveals that most spatial datasets and classes are published by global data providers (such as DBpedia, LinkedGeoData, and Linklion) and cover the whole or large areas of the world. The study of linked data spatial characteristics reveals georeferencing errors or generalizations, including misplaced instances, the "null island" effect (instances located at zero longitude and latitude), the representation of large-area objects (e.g., countries) with points and low population completeness [80] regarding georeferenced instances (e.g., a class about airports contains a random subset of the existing airports). A study of systematic errors and their causes in geographic linked data [84] reveals that about 10% of all spatial data on the linked data cloud are erroneous to some degree. These errors could be minimized if local mapping organizations or agencies participated more actively in the linked data domain since they usually possess complete and high-quality spatial datasets. Some reasons that may prevent their engagement with linked data could be the absence or immaturity of linked data publishing tools and the subsequent high barriers for publishing spatial linked data. One of GeoLOD's goals is to provide an easy-to-use tool that could help users, who are not linked data experts, to get familiar with the linked data landscape and thus to lower the barrier for data publishing. As the usability study indicates, users from the geospatial domain are positive about adopting GeoLOD; however they would like a more user-friendly interface regarding the explanation of terms unknown to them.

GeoLOD includes three innovative features regarding dataset interlinking: (a) a complete list of recommendations for pairs of classes that may include related instances, (b) an on-the-fly recommender for uncatalogued SPARQL endpoints and non-RDF spatial datasets, and (c) automatic generation of SILK and LIMES configuration files. These features help users to discover links between related instances, thus fulfilling the fourth linked data principle, which suggests the establishment of links between related instances so that users can discover related things. In the results, we showed the benefits of employing GeoLOD Recommender as a preparatory step for link-discovery processes. It recommends pairs of classes with 55.98% probability to contain link recommendations between class instances, using a basic link specification in LIMES, while the corresponding probability for random pairs of linked data classes is 6.88%. Furthermore, it reduces the search space for

looking in the Web of Data for candidate classes that can be used as input in link discovery processes 70 times.

We conclude the paper by pointing the future work on GeoLOD. Firstly, the user interface can be improved in terms of providing more help to the users. The catalog can be populated with more content, including spatial datasets that are provided through RDF dumps, listed in other data catalogs (such as LOD Laundromat), using other well-known spatial vocabularies and expressing instance location with line or polygon geometries in various coordinate reference systems. The on-the-fly recommender can be extended to support SPARQL endpoints that use additional spatial vocabularies (other than W3C Basic Geo) and additional spatial data formats, such as the Web Feature Service (WFS) [85]. We plan to take action and conduct experiments to fine-tune the recommendation algorithm's filtering and thresholds criteria and further reduce its overall execution time. Other ideas include the involvement of GeoLOD users so as to provide feedback about "good" or "bad" recommendations and the exploitation of SILK/LIMES web services for instant instance links recommendations.

**Author Contributions:** Conceptualization, V.K.; methodology, V.K.; software, V.K.; validation, V.K. and M.V.; formal analysis, V.K. and M.V.; investigation, V.K.; resources, V.K.; data curation, V.K.; writing—original draft preparation, V.K.; writing—review and editing, V.K. and M.V.; visualization, V.K.; supervision, M.V.; project administration, M.V.; funding acquisition, V.K. All authors have read and agreed to the published version of the manuscript.

**Funding:** This research is being supported by the funding program "YPATIA" of the University of the Aegean.

**Institutional Review Board Statement:** Not applicable.

**Informed Consent Statement:** Not applicable.

**Data Availability Statement:** Data supporting reported results can be found at GeoLOD website at http://geolod.net/ accessed on 16 April 2021.

**Conflicts of Interest:** The authors declare no conflict of interest.

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
