# Peer review of "GeoLOD: A Spatial Linked Data Catalog and Recommender"

_2504-2289, doi:10.3390/bdcc5020017_

Round 1

Reviewer 1 Report

GeoLOD: A Spatial Linked Data Catalog and Recommender

The paper presents GeoLOD, a web catalog for spatial linked datasets. GeoLOD contains details pertaining to the metadata of geospatial linked datasets. Currently, GeoLOD contains metadata for 79 geospatial datasets. Additionally, GeoLOD offers a geospatial-based recommender for similar dataset as well as geospatial classes that may contain related spatial instances. The recommendations can be automatically exported as configuration files for the two link discovery frameworks LIMES and SILK. GeoLOD provides also a web-based user interface and an API for automated content consumption.       

After the introduction section, the authors present the main related work for dataset description, Catalogs and recommenders. Then, they go through the technical details of implementing the GeoLOD catalog and recommender. Finally, the author discussed the statistics of the GeoLOD datasets and presented an evaluation for their recommender.

In general, The paper is well written in general and has enough details and figures that ease the reading of the paper.  In the following I will present my main concerns about the paper:

  • Given that the presented paper is a system paper, I expected to see a standard system usability study [1]. Such study would assess the usability of GeoLOD by different users.
  • The authors took into concern only deduplication tasks (e.g., using owl:samAs)and not investigated more geospatial-related linking, such as finding topological relations [2] and proximity-based linking [3].
  • Concerning the GeoLOD content update rate of 2 months, I think this update rate is too slow given the amount of datasets that appear every day. Also, I would like to know whether or not the updating process is automatically done. In case it is an automatic process, why not implement a continuous 24/7 update process? 
  • The authors did not give a proper introduction of the link discovery problem nor to the link discovery frameworks LIMES [4] and SILK [5].

Other comments:

  • DBPedia → DBpedia
  • Use Latex \texttt{} for predicate names such as “\texttt{owl:sameAs}” instead of just ‘owl:sameAs”
  • In many places, there is an unnecessary space between the word and \footnotes{}. Example line 5 page 5.
  • Figure could be in a better quality if generated as a vector image (e.g., using Latex tikz or simply exported as a pdf file).
  • End of page 6: “The void:classPartition predicate contains the list of datasets spatial classes, …’ → “The void:classPartition predicate contains the list of datasets’ spatial classes, ...”
  • Use \emph{} or \texttt{} to distinguish URI and well known names within text like “node.js”
  • Table 4, 5 and 6 should be centered
  • Using the web UI, I was not able to go to view presented by Figure 4
  •  Caption of Table 3: “N/A denotes that the number could not retrieved.” → “N/A denotes that the number could not be retrieved.”. Also, how come that you were not able to retrieve such numbers, given that you already were able to access the SPARQL endpoints and identify that it contains the spatial classes?
  • Column 5 in Table 6 is empty, which make it hard to get the related claims in the discussion
  • Reference 26 exceeds the line width

[1] Lewis, James R., and Jeff Sauro. "The factor structure of the system usability scale." In International conference on human centered design, pp. 94-103. Springer, Berlin, Heidelberg, 2009.

[2] Sherif, Mohamed, Kevin Dreßler, Panayiotis Smeros, and Axel-Cyrille Ngonga Ngomo. "Radon–rapid discovery of topological relations." In Proceedings of the AAAI Conference on Artificial Intelligence, vol. 31, no. 1. 2017.

[3] Sherif, Mohamed Ahmed and Ngomo, Axel-Cyrille Ngonga. "A Systematic Survey of Point Set Distance Measures for Link Discovery." Semantic Web Journal (2017). 

[4] Ngonga Ngomo, Axel-Cyrille, Sherif, Mohamed Ahmed, Georgala, Kleanthi, Hassan, Mofeed, Dreßler, Kevin, Lyko, Klaus, Obraczka, Daniel and Soru, Tommaso. "LIMES - A Framework for Link Discovery on the Semantic Web." KI-Künstliche Intelligenz, German Journal of Artificial Intelligence - Organ des Fachbereichs "Künstliche Intelligenz" der Gesellschaft für Informatik e.V. (2021). 

[5] Volz, Julius, Christian Bizer, Martin Gaedke, and Georgi Kobilarov. "Silk-a link discovery framework for the web of data." In Ldow. 2009.

Reviewer 2 Report

With the premise that I am not an expert on "linked data", the paper is quite interesting as a presentation of new developments of GeoLOD, but the writing style must be severely revised.

  • The paper massively abuses of the english possessive. It is full of expressions like "user's need" or "datasets' content" and many others. In my knowledge this should be avoided in scientific papers
  • There's a misleading usage of semicolons while should be used a simple comma
    • In the abstract: [...] dataset descriptions in GeoVoID, a spatial dataset  [...]
    • in Introduction: [...] There scenarios are covered in GeoLOD, a web catalog [...]
  • Abstract and Introduction are quite hard to read as they introduce a lot of names that are partly explained in the section 2 on Related Works.
    Nonetheless there are
  • Some "commercial" names are not coherently reported in the paper: DataHub is referred both as datahub.io or Datahub or datahub (the latter is quite confusing since it seems it is a generic concept), while it should be always DataHub.
  • Moreover other "commercial" names are wrongly reported in lower case:
    • europeana instead of Europeana
    • datagov instead of Data.gov
  • On the contrary some generic concepts are reported in capital letters while should be lower case unless they specify a specific tool or product. So while "Catalog" is fine in section 6 when referring the catalog module of GeoLOD in the rest of the paper should be country, search engine, recommender, catalogweb catalog, dataset catalogs, dataset recommenders, bounding box, link discovery framework, and so on.
    If the intent of the authors is to emphasize some terms I suggest the use of italics or quotes or angular brackets or another form but not the capital letters that is very misleading.
    The case of "Link Discovery", that is often reported, is very enlightening: I had to read 'til section 3 before clarify that is not some-kind-of-unknown-to-me software product instead of a generic task and should be always reported, from the very beginning, as "link discovery process (or activity, or task)
  • In 2.2 Dataset catalogs there's a very bad sentence composed by a too long sequence of "ands": [...]statistics about the number AND .... AND ... AND .... AND ... [...]  
  • Some acronyms such as LOV (Linked Open Vocabulary) should be completely reported at their first occurence.  
  • Links, properties, relations, predicates in the RDF form such as "owl:sameAs" or "rdf:type" should be reported in quote or some other ways to distinguish them from normal writing and should be always explicitated: what are, just to name a few,  "owl:sameAs" (a relation?), "owl:EquivalentClass", "dbo:spokenIn", "void:class"? 
    For instance instead you clarify quite well that "rdf:type", "geo:long" and "geo:lat" are "predicates".  

To clarify:

  • In section 1 Introduction, I suppose you mean "38 billions indexed triplets" or with "B" do you mean "bytes" ;-)
  • In section 1 Introduction in the list of scenarios at point 2: I understand that with "linked data provider" you mean a real person since you use "his/her"... I thought it should be a generic provider like a data portal
  • In section 3.2
    I cannot understand on the base of which metric the algorithm ranks the output and who decide the metric?
    Then it is not clear what GeoLOD does if applying the algorithm using only PD, PMI and PHI combined together or exploits the output of the algorithm and then add some other constraints.
    Let's say that the final part of 3.2 section should be revised in order to be more clear.  
  • In sections 5.3, what do you mean with "cleaned label" and "cleaned instances labels"?
  • In section 6 Discussion:
    I can't understand if local mapping organizations should be more active or should be more involved in the linked data domain.
    I confess that "were" is quite confusing to me, but I confess could be my fault.

Corrections:

  • 3.1.2 Data Collection: [...]we retrieve for each dataset (I suppose) its spatial classes [...]
  • there's a typo on "Continent" in Table 4

Suggestions:

  • In sections 3.2 the condition
    PD>0.90 and PMI>1 and PHI>0.02
    suggests that the magnitude of PHI coefficient is much smaller than the others, so instead of summing the 3 coefficients together to get the combined metric, would it be better to multiply them?  
  • In sections 5.3 should be reminded what PD, PMI asnf PHI are, or maybe it would be useful to reference and the similar condition reported in section 3.2
  • At the end of section 5.3, maybe a table reporting the final note data could be useful

Reviewer 3 Report

The authors presented a web catalog of spatial Linked Data datasets and classes. They accompanied the tool with a recommender capable of providing datasets and classes that may contain related spatial instances, generating a ranked list calculated based on a methodology introduced in another work. The proposal allows users to perform Linked Data search, taking into account the spatial characteristics of datasets. Users can perform a search in datasets and classes titles and descriptions, and also by selecting a geographical area through the provided world map. Alongside the web interface explorable through a dedicated website, the authors provided the tool with a set of REST API which can be used by software agents to perform the same operations.

The authors' proposal represents a valuable solution, addressing a current issue in the domain of Geographical Linked Data. However, the overall presentation of the paper lacks the description of many terms and methodologies, making the document not self-contained. The authors do not provide sufficient basis to clarify the involvement of some terms within the presented tool. Terms such as “ASK Query”, “QuadTree”, the characteristics of some fundamental datasets used as a source for this work, are all taken for granted by the authors, which makes the reading very tough. For this reason, I suggest the authors provide a full rewrite of all sections, especially section 3, describing in a more suitable way all the terms and methodologies involved in the proposal.

Furthermore, the authors provided a section where they evaluate the quality of the tool’s recommendation. In this section, the actual procedure applied for the evaluation is not clear either, the authors based the accuracy score on the relation between the recommended classes and the instances found by the LIMES Link Discovery framework. This relation is not justified in a proper way, and for this reason, the Relevant percentage presented is not fully comprehensible and valid. I suggest the authors to apply other evaluation methods for their recommender, in particular, other studies have proved that the adoption of Functional Dependencies can provide a good methodology for the data quality in RDF datasets (https://dl.acm.org/doi/abs/10.1145/2630602.2630605). To this end, Relaxed Functional Dependencies (RFD) can enhance this kind of methodology, since they allow to deal with approximations in both the comparisons between pairs of attributes or on the coverage over a subset of the entire dataset, which might be extremely useful when dealing with linked geographical or multimedia data. Most of the existing algorithms to automatically discover  RFD from data are capable of discovering RFD with only one of the types of approximations mentioned before, because it has been proven that discovering RFD with both types of approximations is an NP-Hard problem. Nevertheless, a suitable solution relying on heuristics is provided in the following paper:

Caruccio, et al: Mining Relaxed Functional Dependencies from Data, Data Mining and Knowledge Discovery, Springer, Vol. 34, 2020.

For a survey on RFD discovery algorithms see:

Liu J, et al (2012) Discover dependencies from data–a review. IEEE Trans Knowl Data Eng 24(2)

I strongly encourage authors to leverage on these papers to derive their own evaluation method for their problem.

Round 2

Reviewer 1 Report

In general, I am happy with the current manuscript. The paper needs a final proofread specially catching  some punctuation typos. For example, a comma should be always added after "e.g.".

Reviewer 2 Report

The paper has much more improved. Well done.
I've seen my suggestions have been accepted and acutally I've found the reading much more easier and interesting.
I've noticed you also significantly increase the number of pages of your paper, and most of the concepts are clearer now.

I particularly appreciate some introductory phrases at the beginning of some chapters where you sinthetically introduce the concepts you are talking about, so that even a non-expert on the subject (as I am) can more easily understand the concepts. I've seen this at 2.3 and 3.1.1.

Also 3.2 as been rewritten more clearer, and I've understand how you get the final ranking.
By the way, I am afraid there's a little typo, or something else I din't catch: just before of 3.2, when you introduce the example of the 3 rankings summed together, you report 1st for PD, 6th for PMI and 3rd for PHI, and so the combined ranking should be 10, not 9... (or did I miss something?).

To conclude I've found just some minor typos in Intruducion:

  • just before the introduction of the "scenarios list" in should be:
    "there in not a tool that addresses users needs" (as the subject is the "tool").
  • And in scenarios list there's an incongruence in writing style since in bullets 2 and 3 you use "his/her" form, while in bullet 4 you simply use "his" form.

Reviewer 3 Report

The authors have addressed most of the main points of my last review in their new version of the paper. It can be accepted in present form.